## Overview Review

**Cite this article:** van der Ven E, Yang X, Mascayano F, Weinreich KJ, Chen EYH, Tang CYZ, Kim S-W, Burns JK, Chiliza B, Mohan G, Iyer SN, Rangawsamy T, de Vries R and Susser ES (2025). Early intervention in psychosis programs in Africa, Asia and Latin America; challenges and recommendations. *Cambridge Prisms: Global Mental Health*, **12**, e3, 1–15

Early intervention programs; first episode psychosis; low- and middle income countries; task-shifting; stigma; traditional healers

**Corresponding author:**
Els van der Ven;
Email: e.m.a.vander.ven@vu.nl

# Early intervention in psychosis programs in Africa, Asia and Latin America; challenges and recommendations

Els van der Ven[1] ![ORCID], Xinyu Yang[2], Franco Mascayano[3,4] ![ORCID], Karl J Weinreich[1] ![ORCID], Eric YH Chen[4,5], Charmaine YZ Tang[6], Sung-Wan Kim[7,8], Jonathan K Burns[9,10], Bonginkosi Chiliza[11], Greeshma Mohan[12], Srividya N Iyer[13,14], Thara Rangawsamy[12], Ralph de Vries[15] and Ezra S Susser[2,3] ![ORCID]

[1]Department of Clinical, Neuro- and Developmental Psychology, Vrije Universiteit Amsterdam, Amsterdam, Netherlands; [2]Department of Epidemiology, Mailman School of Public Health, Columbia University, New York, NY, USA; [3]New York State Psychiatric Institute, New York, NY, USA; [4]Department of Psychiatry, Li Ka Shing Faculty of Medicine, University of Hong Kong, Hong Kong; [5]Key Laboratory of Brain and Cognitive Sciences, University of Hong Kong, Hong Kong; [6]Department of Psychosis, Institute of Mental Health, Singapore; [7]Department of Psychiatry, Chonnam National University Medical School, Gwangju, Korea; [8]Mindlink, Gwangju Bukgu Mental Health Center, Gwangju, Korea; [9]Department of Psychiatry, University of KwaZulu-Natal, Durban, South Africa; [10]Institute of Health Research, University of Exeter, Exeter, UK; [11]Department of Psychiatry, Nelson R Mandela School of Medicine, University of Kwazulu-Natal, South Africa; [12]Schizophrenia Research Foundation (SCARF), Chennai, India; [13]Department of Psychiatry, McGill University, Montreal, Canada; [14]Prevention and Early Intervention Program for Psychosis (PEPP), Douglas Mental Health University Institute, Montreal, Canada and [15]Medical Library, Vrije Universiteit, Amsterdam, The Netherlands

## Abstract

**Background:** While early intervention in psychosis (EIP) programs have been increasingly implemented across the globe, many initiatives from Africa, Asia and Latin America are not widely known. The aims of the current review are (a) to describe population-based and small-scale, single-site EIP programs in Africa, Asia and Latin America, (b) to examine the variability between programs located in low-and-middle income (LMIC) and high-income countries in similar regions and (c) to outline some of the challenges and provide recommendations to overcome existing obstacles.

**Methods:** EIP programs in Africa, Asia and Latin America were identified through experts from the different target regions. We performed a systematic search in Medline, Embase, APA PsycInfo, Web of Science and Scopus up to February 6, 2024.

**Results:** Most EIP programs in these continents are small-scale, single-site programs that serve a limited section of the population. Population-based programs with widespread coverage and programs integrated into primary health care are rare. In Africa, EIP programs are virtually absent. Mainland China is one of the only LMICs that has begun to take steps toward developing a population-based EIP program. High-income Asian countries (e.g. Hong Kong and Singapore) have well-developed, comprehensive programs for individuals with early psychosis, while others with similar economies (e.g. South Korea and Japan) do not. In Latin America, Chile is the only country in the process of providing population-based EIP care.

**Conclusions:** Financial resources and integration in mental health care, as well as the availability of epidemiological data on psychosis, impact the implementation of EIP programs. Given the major treatment gap of early psychosis in Africa, Latin America and large parts of Asia, publicly funded, locally-led and accessible community-based EIP care provision is urgently needed.

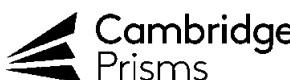



## Impact statement

Early intervention in psychosis (EIP) programs aim to offer evidence-based case management, psychopharmacology and psychosocial support in the early stages that individuals seek help for psychosis. While the number of such programs in high-income countries is steadily growing, much less is known about their existence in low- and middle-income settings, specifically in large parts of Asia, Africa and Latin America. We also have much to learn about the characteristics of regions in terms of economy and healthcare system and the association with the successful implementation of these care models. Overall, few EIP programs are population-based and cover a large proportion of the population presenting with a first psychotic episode. Most programs are single-site programs that have incorporated the philosophy of EIP care but are not scalable and able to reach a high proportion of people with early psychosis at the country level. This review provides an overview of EIP programs in Africa, Asia and Latin America and focuses primarily

on low-and-middle-income countries and those with developing economies. We also discuss the variability in programs according to country-level income in these regions. Based on this review, we describe challenges and practical recommendations to enhance the implementation of early psychosis care in global settings.

## Introduction

Psychotic disorders including schizophrenia are globally prevalent mental disorders that impede social and occupational functioning, quality of life and physical health. The last three decades have seen a paradigm shift in the treatment of psychosis with an emphasis on early intervention and intensive mental health service provision in the early stage of the illness (McGorry et al., 2008). Early intervention in psychosis (EIP) programs is predicated on research demonstrating that a longer duration of untreated psychosis (DUP) leads to poorer outcomes (Chen, 2019; Howes et al., 2021). Several systematic reviews and meta-analyses consistently show a small to modest effect of DUP on symptomatic and functional outcomes in the first year after illness onset (Marshall et al., 2005; Penttilä et al., 2014; Perkins et al., 2005). Most studies included in this work are conducted in high-income countries (HICs). However, an association between longer DUP and a poorer response to treatment and increased levels of disability has also been observed in various low-and-middle-income countries (LMICs) (Chiliza et al., 2012; Farooq et al., 2009; Chen, 2019). Furthermore, there is evidence that DUPs are significantly longer in LMICs (Large et al., 2008), highlighting the need for relevant, effective, culturally acceptable and potentially scalable EIP programs in these world regions (Lilford et al., 2020).

EIP care is usually provided by specialized teams with a reduced caseload compared to conventional mental health services. Teams consist of multiple disciplines generally offering evidence-based intensive case management, pharmacological management and psychosocial support with the goal of enabling outreach and promoting engagement (McGorry, 2015). In addition to clinical outcomes, personal recovery and improvements in occupational, social and personal domains are prioritized (McGorry et al., 2008). Its counterpart in the United States is coordinated specialty care which comprises multicomponent care types of services including several psychosocial and psychopharmacological interventions (e.g., case management, psychotherapy, supported employment and education and family support) that are provided from one team in a coordinated, integrated fashion (Bello et al., 2017). The philosophy for service provision includes concepts that stimulate engagement such as shared decision-making, meaningful peer worker involvement, outreach and culturally competent care (Thomas et al., 2022). Programs are offered during the early phase (typically in the first two to five years) of a psychotic disorder.

The history of the establishment of EIP care varies by context. In the 1980s, studies distinguishing first-episode psychosis (FEP) from more chronic phases of the illness, described the potential benefits of initiating pharmacological treatment early after the onset of psychosis (Kane et al., 1982; Crow et al., 1986). These initial findings led to the inception of the Early Psychosis Prevention and Intervention Centre (EPPIC), an EIP care model, aimed at providing comprehensive services to all individuals with FEP within a large catchment area in Melbourne, Australia (McGorry et al., 1996). Since then, the population-based EIP care model has been implemented in the UK and several Northern European countries, parts of North America and a few East Asian countries. While the nature and effectiveness of these programs have been widely documented, especially in North-Western Europe and East Asia much less is known about EIP initiatives from Africa, Latin America and other parts of Asia.

Hundreds of EIP programs have been initiated worldwide, although the level of intensity, amount of peer involvement and leadership, duration of follow-up and threshold to enrollment varies substantially. Several programs have demonstrated the beneficial effects of EIP care on clinical outcomes compared to care as usual. The OPUS trial in Denmark found positive effects after two years of follow-up, although the benefits of EIP diminished over time (Hansen et al., 2023). A meta-analysis of 10 randomized clinical trials including more than 2000 individuals enrolled in EIP programs in Hong Kong, Mexico, the US and various European countries reported favorable outcomes in multiple domains including involvement in school or work, quality of life and symptom severity (Correll et al., 2018). A challenge of this work is that most RCTs include single-site, small-scale, rather than "real-world", population-based programs (Correll et al., 2018). Single-site programs usually provide access to a highly selective subgroup of people with FEP and often remain inaccessible to disadvantaged communities (van der Ven and Kirkbride, 2018). Furthermore, young adulthood is one of the peak periods of psychosis onset and, as LMIC countries have predominantly young populations, it is not surprising that most people with early psychosis worldwide are to be found in LMIC contexts (Patel et al., 2018; Jongsma et al., 2019). Nonetheless, the vast majority of EIP programs are in HICs.

The primary goal of this narrative review is to provide an overview of EIP programs in Africa, Asia and Latin America, regions classified as "developing economies" by the United Nations (United Nations, 2014) and/or those classified as LMICs by the World Bank classification (World Bank, 2022). These classifications are for the most part overlapping, barring some exceptions (e.g., Chile, which is a developing economy, but not an LMIC). The specific aims are to describe population-based and single-site, small-scale EIP programs in Africa, Latin America and Asia, and to discuss the variability in EIP programs between and within LMICs and HICs in these regions. Lastly, we will discuss existing challenges and provide recommendations to advance the clinical and research field in relation to the implementation of EIP in under-resourced settings.

## Method

The present narrative review aimed to highlight key EIP programs implemented in under-resourced contexts across the globe. To identify relevant programs, we employed a two-tier approach of collecting expert input and conducting a systematic literature search. First, a number of experts (listed as co-authors) in the field were identified based on their expertise on EIP programs in any of the targeted regions (Africa, Asia and Latin America) and invited to contribute their expertise and additional knowledge of local EIP programs, as well as discuss published literature. The authors additionally reached a consensus on shared challenges and future directions for the field of EIP in the included settings. Second, we conducted a systematic search guided by the scale for the quality assessment of narrative review articles (SANRA)(Baethge et al., 2019). We searched in Medline, Embase, EBSCO/APA PsycInfo, Web of Science (Core Collection)

and Scopus databases from inception up to February 6, 2024, in collaboration with a librarian. The following terms were used (including synonyms and closely related words) as index terms or free-text words: "early intervention", "coordinated specialty care", "scalable intervention", "psychotic disorders", "schizophrenia" and "Low- and Middle-income countries". The references of the identified articles were searched for relevant publications. All languages were accepted. Duplicate articles were excluded using Endnote X20.0.1 (Clarivate[tm]), following the Amsterdam Efficient Deduplication (AED)-method (Otten et al., 2019) and the Bramer method (Bramer et al., 2016). The full search strategies for all databases can be found in the Supplement. Two reviewers (EvdV and KJW) independently screened all potentially relevant titles and abstracts for eligibility. If necessary, the full-text article was checked for the inclusion criteria. Differences in judgment were resolved through a consensus procedure. Our inclusion criteria for papers were that they described a program: a) situated in countries with emerging economies or LMICs; b) targeting FEP or recent-onset psychosis; c) aimed at improving the detection or intervention of FEP or recent-onset psychosis. All identified studies were included in the qualitative synthesis. Available information on the length of follow-up, number of sites, target population, EIP program components offered, funding source and delivery personnel was collected and synthesized.

## Results

In total, eight programs in countries with emerging economies and LMICs were identified through the systematic search (Table 1) and one additional program, i.e. situated in Argentina, was included through expert input. The flow chart of the search and selection process is presented in Figure 1. From the EIP programs available, two types of programs can be distinguished. First, population-based programs are generally integrated into a country's mental health care system and accessible to the population at large. Second, some programs exist as standalone, single-site programs that are often early adopters of the EIP model (Maric et al., 2019). Population-based programs are distinctly different in terms of (a) scale, i.e. these programs intend to identify and/or provide care to all new cases of psychosis at a regional or country level; (b) strategic development and implementation, i.e. the development and implementation of programs are based on data that demonstrate the mental health care need for psychotic disorder in well-defined regions; and (c) patient selection, i.e. efforts are made to lower barriers to care and to improve referral pathways to specialized FEP programs.

Few countries across the globe, including HICs, have implemented population-based treatment programs that are available free of charge to individuals presenting with early psychosis. Some exceptions include the OPUS program in Denmark, EPPIC in Australia, EASY in Hong Kong, and the National Health Service Plan in the UK (Hansen et al., 2023; McGorry et al., 1996; Joseph and Birchwood, 2005). Even in high-income settings full universal coverage, including remote, less densely populated areas, is rare. In the selected continents, two countries with emerging economies have attempted to implement population-based programs for early psychosis, Chile and China, which will be discussed separately.

**Table 1.** Overview of characteristics of the selected programs in Asia and Latin America

| | Duration of follow-up (in years) | Number of sites | Target group | Components | Funding | Personnel |
|---|---|---|---|---|---|---|
| Mainland China (686 program) | Not specified | National | Severe mental disorders, including specific FEP component | Pharmacotherapy, hospital care, basic outpatient care | Public | Various mental health professionals |
| Hong Kong (EASY) | 3 | 7 | FEP (age 15–65) | Multi-disciplinary teams, outpatient care, hospital care, stigma reduction | Public | Psychiatrist, case manager, clinical psychologist |
| India (SCARF) | 2 | 1 | FEP (age 16–45) | Pharmacotherapy, psychosocial interventions, case management, psychoeducation, hospital care, multi-disciplinary teams, outpatient care | Research funding | Psychiatrist, psychologist, social worker, employment specialist, other allied healthcare professionals |
| Chile (OnTrack) | 2 | National | FEP (first 6 months) | Early detection, pharmacotherapy, psychosocial interventions, psychological interventions, outpatient care, community support, family interventions | Public | Team coordinator, psychologist, occupational therapist, psychiatrist, nurse |
| Brazil (PEP, UNIFESP-EPM) | 2 | 1 | FEP (< 3 month adequate treatment) | Pharmacotherapy, psychoeducation, family interventions | Unclear | Unclear |
| Brazil (Ribeirão Preto EIP) | 2 | 1 | Psychotic symptoms | Pharmacotherapy, psychoeducation, family interventions, occupational therapy | Research funding | Psychiatrist, nurse, occupational therapist, psychologist |
| Mexico | 0 5–1 | 1 | FEP (age 16–50) | Pharmacotherapy, psychosocial interventions, psychoeducation | Research funding | Psychiatrist, clinical psychologist, family therapist |
| Malawi | Not specified | 1 | Help-seeking individuals with psychosis | Community psychoeducation, referral hotline, mental health services | Unclear | Community mental health care team |

FEP = First Episode Psychosis

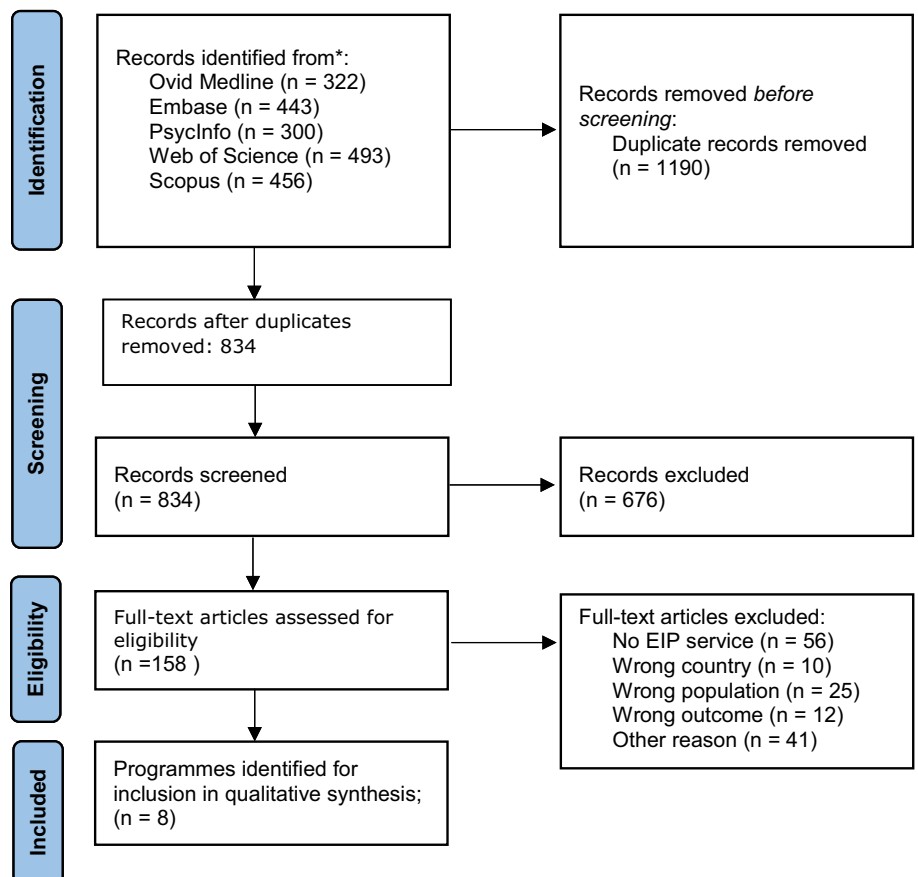

**Figure 1.** Flow chart of study selection.

## South-East and East Asia

### *Population-based programs*

#### *Mainland China*

In China, financial resources, available psychiatric beds and mental health care workers are distributed unequally between urban and rural areas (Liang et al., 2018). For instance, urban centers like Shanghai have access to concentrated resources (Liang et al., 2018), while most rural counties did not have any psychiatric beds up to 2012, making mental healthcare relatively inaccessible to the rural population (Chang and Kleinman, 2002; Patel et al., 2016; Xiang et al., 2018). In response to these disparities, the Central Government Support for the Local Management and Treatment of Severe Mental Illnesses Project, in short, "686 program", aims to close the coverage gap between urban and rural areas by reducing reliance on specialist psychiatric hospitals and integrating mental health care into the general healthcare system (Good and Good, 2012).

The focus of the program is mainly on individuals with psychotic disorders, with the project aiming to provide screening, identification, treatment and monitoring, free of cost if necessary (Liang et al., 2018; Liu et al., 2011; Ma, 2012; Xiang et al., 2018). As part of a national health program, it registered 5.4 million individuals with severe mental illness by 2015, three-quarters of which were diagnosed with schizophrenia (Xiang et al., 2018). The program set out to integrate resources from hospitals, community services and police and has, according to not independently verifiable information published by the Chinese government, provided services including prevention, treatment and rehabilitation to 88.7% of patients (Liang et al., 2018; Xiang et al., 2018). However, a lack of

clear guidelines for funding allocation and regional differences in service provider participation has led to disparities in implementation and, subsequently, in care between regions (Liang et al., 2018).

Importantly, in its current state, the 686 program is not a universal EIP program in a common sense but rather offers the basic provision of minimal outpatient services. Unlike the route taken in Chile, the 686 program appears to follow a different philosophy. Its focus seems to be more geared toward keeping social harmony and preventing potential violence by individuals with severe mental illness, rather than on recovery, shared decision-making and personal needs (Liang et al., 2018; Liu et al., 2011; Xiang et al., 2018). Concerns regarding the implementation of the intended human rights-based approach remain, as guidelines leave room for interpretation and subsequently, progress toward this goal has been slow. Issues such as providing adequate patient care and the potential misuse of the mental health system patient data by security services remain largely unaddressed (Jiang et al., 2018; Shao et al., 2015; Xiang et al., 2012; Yao et al., 2022; Good and Good, 2012; Liang et al., 2018; Patel et al., 2016).

Significant obstacles to the effective implementation and success of the program such as a focus on treatment delivery exclusively via hospitals, the stigma surrounding mental health, and an inadequately educated workforce have persistently remained (Liang et al., 2018; Patel et al., 2016). Additionally, while individuals living in poverty receive free treatment and national health insurance does cover mental health expenses, fees potentially remain a barrier, as reimbursement is often incomplete and only covers basic treatments, such as medication. Despite the program's ambitious goals,

a lack of comprehensive, accessible and high-quality data has made systematic evaluation of the program's implementation, fidelity, reach and goal achievement impossible (Liang et al., 2018; Patel et al., 2016; Zhou and Xiao, 2015).

### Hong Kong and Singapore

There are several universal early adopters of EIP programs in high-income regions in Asia, including Singapore and Hong Kong. These sites share some societal features, such as generally being relatively affluent overall while still struggling with high levels of stigma and low service resources for mental health. There are also important differences between communities, particularly in the way public services are funded, which have important implications for early detection work. Clinician-researchers from the region have initiated a professional network (the Asian Network for Early Psychosis) which has been meeting regularly for the sharing of ideas, resources and experience (Asian Network of Early Psychosis Writing Group, 2012).

Early Psychosis programs in Singapore (Early Psychosis Intervention Program, EPIP) and Hong Kong (Early Assessment Service for Young People with Psychosis, EASY), both started in 2001 (Verma et al., 2012a, 2012b; Tang et al., 2010). Both programs are innovations situated in conventional public-funded mental health programs where the health care provisions for the entire population are considered (rather than a predominantly fee-for-service system) (Verma et al., 2012b; Tang et al., 2010). While EASY is entirely free of charge, EPIP provides care at a heavily subsidized rate. Both EPIP and EASY started as population-based programs focused on young people (Verma et al., 2012a, and 2012b; Tang et al., 2010; Chan et al., 2018). They adopted specialized multidisciplinary teams with case managers being at the core of the service, providing continuous community and tertiary-level support for patients with FEP covering the first two years after diagnosis (this was later extended to three years) (Chen et al., 2015). Both programs adopted public awareness campaigns to increase community awareness of psychotic disorders to reduce their associated stigma, in the hope that this would reduce the DUP in the population (Verma et al., 2012; Tang et al., 2010; Chan et al., 2016; Chen, 2019). The EIP programs endeavor to use protocol-based practices to monitor and improve functional outcomes in the 2–3 years covered by the service, as well as more lasting outcomes in the longer term (Verma et al., 2012; Tang et al., 2010).

In Singapore, EPIP was followed by the development of an upstream program targeting at-risk mental states (Support for Wellness Achievement Program, SWAP), as well as a community youth mental health program (CHAT, formerly known as Community Health Assessment Team) (Tay et al., 2014; Chan et al., 2019; Chua et al., 2019; Lee et al., 2019; Harish et al., 2021). To cater to those with early-onset psychosis, the EPIP service, which was originally for those aged 16 to 40, was extended to people aged 12 to 40; with the duration of service provision extending from two to three years (Hui et al., 2020). In Hong Kong, a successful pilot study to provide case management to patients over 25 led to the extension of EIP service from the 15–25 age range to the entire adult age range (15–65) (Hui et al., 2013; Hui et al., 2022). An RCT of 3 years vs 2 years of case management showing the superiority of longer follow-up has led to the provision of three years rather than two years of case management (Chen et al., 2015). In Singapore, the nationwide service was based at a single tertiary mental health institution (the Institute of Mental Health), whereas in Hong Kong, the service was distributed across seven regional service clusters (Tan et al., 2019). This has implications for staff

competency-building (ability to engage, to assess and to intervene) which was more continuous in Singapore than in Hong Kong. As a result, the passing on of experience and information between successive generations of staff was more challenging for Hong Kong than for Singapore. On the other hand, the service programs in Hong Kong were more closely linked to university research programs, providing opportunities for data acquisition and follow-up studies. Importantly, the implementation of EIP may have positive trickle-down effects on other mental health services. The EIP service in Hong Kong was the first (1) to use case management, (2) to use extensive service evaluation, and to (3) spearhead community public awareness and anti-stigma campaigns. Case management developed in EIP served as a model for later case management in generic services. Similarly, the use of more extensive outcome measurement in programs was adopted by other mental health services. Anti-stigma campaigns focused on psychosis are also expected to benefit other mental health conditions (such as depression) as they are generally regarded as less stigmatizing than psychosis. However, a measurable, direct impact of trickle-down effects is missing.

### South Korea

Various university-initiated EIP programs are operated in Seoul, Jeonju and Gwangju in South Korea (Kwon et al., 2012; Na et al., 2015; Kim et al., 2020). For example, a community-based EIP service for youth in Gwangju (Mindlink) aims to detect mental illness in young people early and provide comprehensive multidisciplinary interventions (Kim et al., 2020). Many young people with distressing mental illnesses, and their family members are voluntarily seeking early psychiatric treatment despite the major associated stigma (Kim et al., 2020). The model is being taken up in other areas in Korea. Although its coverage is not yet at the national level, the program has been scaled up rapidly from the first site in Gwangju in 2012 up to eight sites in 2023.

### Single-site programs

The publicly funded, population-based EIP programs in Hong Kong and Singapore can be contrasted broadly with initiatives in some other areas in Asia (for example in Japan), where funding is more inclined toward a fee-for-service system (Mizuno et al., 2012). In the latter context, it is more challenging for service providers to argue for investing in early detection and community intervention, as this may compromise the "income" for a service (Takamura et al., 2011; Chan et al., 2019). EIP programs can effectively reduce the need for hospitalization of FEP and hospitalization could be one of the main sources of income for the service (Takamura et al., 2011; Chan et al., 2019; Chan et al., 2020). In these situations, there are fewer incentives to provide early detection programs (Takamura et al., 2011). Instead, many initiatives have been associated with university research and pilot programs. In many cases, the focus has been on the clinical high risk rather than the FEP population.

In Japan, the main service platform for psychiatry is hospital-based, and many of the hospitals are private (Mizuno et al., 2012). Psychosis has been heavily stigmatized. A change of the Japanese name for schizophrenia to "integration disorder" may have partially reduced stigma (Maruta and Matsumoto, 2018). Schizophrenia is now seen as only marginally more stigmatizing than depression or a cultural nonpathological idiom of distress (hikikomori; DeVylder et al., 2020). This indicates a slight shift in the perception of schizophrenia as a syndrome, rather than a

disease based on a brain vulnerability, and moves away from prior associations with criminality (Maruta and Matsumoto, 2018).

A notable example of an early detection program was the Il Bosco youth engagement center operated by the Toho University team in Tokyo, which provided engagement for at-risk mental states as well as FEP (Mizuno et al., 2012; Nemoto et al., 2012). In Taiwan, the EIP program has been associated with a robust research program targeting the clinical high risk as well as the first episode psychosis at the National University of Taiwan (Liu et al., 2010). In China, apart from Hong Kong, there have been single-site programs as well, mostly related to academic centers. For example, the multi-center first episode psychosis project including centers in various cities such as Beijing and Shanghai provided a good starting point for specialized EIP care (Han et al., 2014). It is important to note that this list is more by way of example rather than exhaustive.

## South Asia

South Asia, a diverse and rapidly growing southern region of Asia, includes Afghanistan, Bangladesh, Bhutan, India, Maldives, Nepal, Pakistan and Sri Lanka (Trivedi et al., 2007). More than 23% of the world's population lives here, and approximately 150–200 million people suffer from mental disorders, which are often under-addressed due to several common challenges (Trivedi et al., 2007). There are only four countries with national mental health policies: India, Pakistan, Nepal and Bhutan (Trivedi et al., 2007). However, actual mental health services are nonexistent or very basic due to a severe lack of resources and trained staff to diagnose, treat and prevent mental health problems. There are no population-based EIPs and very limited single-site programs.

## *India*

In South Asia, it is only India that recognizes the importance of early intervention in mental health reform (Gupta and Sagar, 2021). Through the National Mental Health Policy 2014, National Health Policy 2017 and Mental Healthcare Act 2017 (MHCA). India's National Adolescent Health Program promotes screening and early detection of health problems, including mental health, at schools and providing access to care (Barua et al., 2020). However, much more needs to be accomplished as a range of systemic barriers, and factors related to political, contextual, organizational and community participation limit the scope and implementation of the various policies and programs in the country (Singh et al., 2015).

Among the very few early psychosis programs is the first episode psychosis program at the Schizophrenia Research Foundation (SCARF) located in the southern state of Tamil Nadu (Rangaswamy et al., 2012). SCARF is a nongovernmental organization and a World Health Organization collaborating center. In 2003, SCARF's FEP program began under the aegis of a research collaboration with the Prevention and Early Intervention Program (PEPP-Montreal), affiliated with McGill University in Canada (Rangaswamy et al., 2012). Through this collaboration, which was funded through two National Institutes of Health (NIH) grants, a multidisciplinary, EIP program with an embedded research infrastructure was set up at SCARF and a prospective longitudinal study comparing multiple outcomes over a two-year follow-up among persons followed at SCARF (N=168) and PEPP (N=165) in Montreal and their families was conducted (Iyer et al., 2010; Malla et al., 2020). SCARF's EIP program has since been sustained and has

participated in additional services research projects including the Warwick-India-Canada project funded by the U.K.'s National Institutes of Health Research (Singh et al., 2021).

The program serves those between the ages of 16–45 years who meet the criteria for a primary DSM-IV-TR diagnosis of either schizophrenia-spectrum psychotic disorder or affective psychosis and have not received antipsychotic medication for more than 30 days since the onset of psychosis (Malla et al., 2020). Awareness programs are carried out as part of SCARF's activities in schools, colleges, corporate offices, print and visual media. The age criterion had an upper limit of 35 years during the Canada-India NIH-funded study, but since then has been revised to 45 years to be more inclusive. Patients in the program are followed for two years by a multidisciplinary team and receive a wide range of psychosocial and medical services including assertive case management, family psychoeducation, antipsychotic medication and as indicated, other individual and family psychosocial interventions. Upon completion of the two-year follow-up, users are discharged to the general outpatient program at SCARF, in which they have access to a variety of services like supported employment, psychosocial rehabilitation or vocational training. While the program is informed by international EIP guidelines, adaptations to enhance its fit to the local context and preferences were integrated such as focusing on household chores during cognitive remediation. (Rangaswamy et al., 2012).

The Canada-India collaboration highlights numerous valuable insights. On average, patients in Chennai, India were in their mid-twenties, had completed high school, and were living with their families (Malla et al., 2020). Most were diagnosed with schizophrenia-spectrum disorder and had an average DUP of 32.82 weeks (median = 11.8 weeks; range = 0.29–518.71 weeks) (Malla et al., 2020). The male-to-female ratio (49% men,51% women) was not as skewed as is typically the case in HIC cohorts, a finding also borne out by the INTREPID epidemiological cohort study in India (Morgan et al., 2022). After accounting for demographic characteristics and other pertinent covariates, negative (but not positive) symptom outcomes were better in Chennai compared to Montreal (Malla et al., 2020). Interestingly, a higher proportion of Chennai patients (49% compared to 17% in Montreal) went off (and stayed off) antipsychotic medication over the two-year course, with no differential impact on clinical and functional outcomes in the "off" (compared to the "on medication") group in Chennai (Malla et al., 2020). Additionally, Indian families were engaged with treatment consistently (nearly every month) at a high level, while family engagement decreased over time in Canada. This decrease is likely due to a number of factors including the patient, rather than the family, being seen as the primary unit of therapeutic attention in HIC contexts like Montreal (Iyer et al., 2022). This is important as early family support was associated with improved negative symptom outcomes (Malla et al., 2020), suggesting that higher family involvement may be contributing to better outcomes in Chennai (Iyer et al., 2022). Comparative analyses of additional patient and family outcomes and experiences are ongoing and suggest that contexts may have differential effects depending on the type of outcome. Disconcertingly, four persons died over the two-year follow-up in the India cohort (three by suicide) compared to none in Montreal (Malla et al., 2020). The study has also supported the development of several patient-reported and other tools and pushed attention toward hitherto neglected but important dimensions as we build EIP programs in global contexts such as patient and family experiences of feeling supported by the treatment team,

and patient, family and treatment provider perceptions about sharing responsibility for addressing the needs of those with mental illness.

Through the Warwick-India-Canada study, SCARF supported the creation of a protocol for early psychosis programs in LMIC settings and collaborated around the creation of an EIP program in a tertiary care setting (All India Institute for Medical Sciences) in New Delhi, India (Singh et al., 2021). Established early intervention programs are not present in other South Asian countries and this could be attributed to mental health not being a priority for many governments in the region hence the funds allocated are insufficient (Thara and Padmavati, 2013).

### Pakistan and Nepal

There are virtually no EIP programs in these regions, but there are some studies providing potentially valuable information for developing such care models. For instance, studies have provided insights regarding pathways to care. Individuals with psychosis commonly seek help from traditional and faith healers, with between 15% in Pakistan (Naqvi et al., 2009), 25% in the SCARF cohort in India (MacDonald et al., 2023) and up to 59.4% in Nepal (Dhungana & Ghimire, 2017; Gupta et al., 2021) consulting traditional healers as a first point of contact. Nevertheless, the effectiveness of traditional healers in treating psychosis has not been convincingly demonstrated (Nortje et al., 2016). While consultation with traditional healers as a first point of contact has been associated with increased DUP (Gupta, Grover, et al., 2021; Lilford et al., 2020), there may also be important advantages of including this group in standard models of care. Findings from the Program for Improving Mental health Care (PRIME) conducted in five countries including Nepal and India, suggested that contact with traditional healers may be incorporated into existing systems of mental health care as additional culturally adapted supports (Mendenhall et al., 2014). It has been argued that consulting traditional healers could be conceptualized as a form of social support, and there is some evidence pointing toward traditional healers having beneficial effects on common mental disorders and on individuals' quality of life (Naeem et al., 2015; Nortje et al., 2016). A collaborative approach between traditional healers and mental healthcare workers could subsequently be promising, particularly when respective strengths regarding western and local concepts of healing and wellness are integrated (Gureje et al., 2015). Enabling patients to select traditional healers as an adjunctive treatment that aligns with their understanding of illness could allow treatment to benefit from patients' expectations (Gureje et al., 2015; Koss, 1987; Naeem et al., 2015) and play a role in alleviating stigma by providing culturally meaningful treatment and facilitating community reintegration (Angermeyer et al., 2016). Since pathways to care in LMIC frequently start with consultation of traditional healers (Dhungana & Ghimire, 2017; Gupta, Grover, et al., 2021; Gupta, Joshi, et al., 2021; Hashimoto et al., 2015; Lilford et al., 2020; MacDonald et al., 2023), their education and training regarding referrals, integration into care networks and involvement in task shifting could be beneficial (Gureje et al., 2015; Padmavati et al., 2005; World Health Organization, 2013).

In Bangladesh, an ethnomedicinal survey of plants used to treat schizophrenia by traditional medical practitioners was conducted. Compounds that could potentially have beneficial effects were present in a number of commonly used plants (Ahmed & Azam, 2014), indicating the potentially beneficial role of local cultural practices for EIP programs.

Overall, while the importance of mental health care is gradually improving in some South Asian countries, constraints to access, availability and affordability of care primarily due to poor financial resources remain the primary challenge. Because of lack of funding, of trained mental health professionals, inpatient, emergency and crisis facilities, and of psychotropic medications, Western and East Asian EIP models of care have been criticized as being difficult to implement in LMICs in South Asia. For such criticisms to be addressed and to increase the allocation of funds toward mental health, more evidence-based data is urgently needed.

## Latin America

### Population-based programs

#### Chile

Since 2005, Chile has had a universal-access, population-based, program that prioritizes early diagnosis and treatment access for people with FEP (Mascayano et al., 2022). The schizophrenia treatment program was launched as part of a larger healthcare policy reform known as the program of Explicit Health Guarantees, which is regulated by the Chilean government. It is aimed to provide quality health services targeting a selected list of prioritized health conditions (Le et al., 2022). Although the former Chilean program does not align with the typical EIP program framework in terms of staffing, caseload management and other core program principles, it is in the process of adopting these components in order to transition into a standard EIP program. By doing so, it is the first country in the Global South to implement specialized EIP programs at such a large scale.

Individuals identified as having a FEP diagnosis or suspected FEP are entered into the Chilean registry and are entitled to free evaluations and potential treatment (Le et al., 2022; Gaspar et al., 2018). Special attention is given to the first 6 months to determine diagnosis. The goal is to facilitate the identification of people with FEP, the diagnostic process, as well as timely access to care so the system can better serve people's needs (Minoletti et al., 2021). In addition to pharmacological treatment, individuals diagnosed with FEP are entitled to psychosocial interventions, psychotherapies, or certain forms of community-based services (Minoletti et al., 2021). The national schizophrenia treatment program has played a crucial role in improving access to mental health care for schizophrenia patient populations (Minoletti et al., 2021). The identification of early psychosis in Chile is a crucial aspect of mental health care, and it currently relies on an extensive network of primary care clinics and community mental health centers. However, this network does not always function as an integrated system of care, which most likely leads to gaps and delays in the identification and treatment of early psychosis.

However, the programs promoted by the FEP policy and provided at outpatient clinics usually do not correspond to the kinds of services supported by current evidence. For instance, although over 80% of FEP clients in Chile receive medications, only 40% receive other important services such as support for education and employment, family counseling and peer support (Alvarado et al., 2012). Moreover, when these other services are offered, they tend to be ad hoc because most providers are not trained in evidence-based, recovery-oriented approaches. All are critical areas that can be addressed by components commonly integrated in specialized FEP programs.

Since 2019, extensive efforts have been made to scale up these programs by implementing OnTrack Chile, a FEP program derived from the well-known OnTrack New York program, including community-based, recovery-oriented social interventions (Mascayano et al., 2019). OnTrack Chile, a U.S.-funded effort, offers an adapted version of OnTrack New York, a large program currently being implemented across New York State and the US (Mascayano et al., 2019). OnTrack Chile offers a variety of recovery-oriented, person-centered services including Cognitive Behavioral Therapy for psychosis, psychiatric medications and supported education and employment (Mascayano et al., 2019). The effectiveness (e.g., personal recovery, functioning) and implementation (e.g., adoption, acceptability) of OnTrack Chile are being assessed in a cluster, hybrid type 1 RCT (n=300), in several regions of Chile (Mascayano et al., 2022). Initial qualitative analyses show that participants (i.e., clients and providers) have expressed enthusiasm and support for OnTrack's principles of care. However, some participants reported reticence, citing the cultural norm that patients and their families typically expect to have passive roles in treatment. Participants also highlighted numerous challenges, including spatial and financial constraints that should be addressed (Le et al., 2022; Mascayano et al., 2022).

Importantly, the Ministry of Health had already incorporated the recovery model before the inception of OnTrack Chile (Ministerio de Salud, 2018). In terms of reach, the Ministry of Health aims to improve detection and referral in primary care, particularly focusing on rural communities and migrants. However, the extent to which these efforts are successful is unclear.

### Single-site programs in other Latin American countries

In a recent analysis, Kohn et al. (2018) reported that the treatment gap for severe mental disorders in Latin American countries was 69.9% and 74.7% for severe to moderate disorders. Moreover, the treatment gap for substance use disorders was 83.7% compared to 69.1% for North America (Kohn et al., 2018). Access to psychotropic medication remains an issue in a large proportion of Latin American countries. For instance, antidepressants and antipsychotics were available in less than 20% of health centers and small health clinics in Peru (Hodgkin et al., 2014). Moreover, psychosocial, community-based treatment for people with mental disorders is unavailable in many settings (Pan American Health Organization, 2013). Policy changes in general health care sometimes explicitly give low priority to mental health care, and mental health budgets are often much lower than optimal.

As noted in a previous literature review (Aceituno et al., 2020), EIP programs are usually not offered in Latin America. With notable exceptions, such as Chile, Mexico and Brazil, where mental health care has been substantially strengthened in the last decade, outpatient and community care for early psychosis, including EIP programs, is largely undeveloped. Even though recovery-oriented approaches have been increasingly adopted in HICs, such programs are rarely offered in the region and are not yet integrated into universal healthcare services. The number of EIP programs between 2011 and 2020 has remained practically the same (Brietzke et al., 2011; Aceituno et al., 2020).

Nonetheless, progress has been made. Hospital-based and research-funded initiatives have reported the implementation, feasibility and appropriateness of different EIP programs in Brazil (Fabri Cabral & Chaves, 2009), Chile, Mexico (Valencia et al., 2012) and Argentina (Padilla et al., 2015). Recently, Aceituno et al. (2020) noted that seven initiatives to improve EIP care can be found in the

region. With the exception of the national Chilean programs, most operate at a very small scale and have not been thoroughly assessed from both effectiveness and implementation perspectives. We focus here on a limited number of programs as examples, including the "Psychosis Episode Program of the Federal University of Sao Paulo (UNIFESP-EPM)" and "OnTrack Chile", given our familiarity with these initiatives and their magnitude and public health influence. Moreover, these two initiatives have different trajectories as UNIFESP-EPM is a well-established program, compared to OnTrack Chile, which is currently implemented in a large cluster RCT.

When UNIFESP-EPM was initiated in 1999, it became one of the earliest EIP programs not only within the nation but in the Latin American region (Aceituno et al., 2020). Targeting FEP patients referred from psychiatric emergency services not restricted to a defined catchment area, UNIFESP-EPM was designed as a comprehensive outpatient treatment program that combines applications of low-dose antipsychotics, support groups, as well as psychoeducational multi-family group intervention (Chaves, 2007). Between 2002 and 2003, 63 first-episode patients were included in the program. Over half of the patients were male, with an average age of 23. Findings from qualitative interviews have demonstrated feasibility and showed that the program was well-accepted by family members and caregivers (Chaves, 2007). More recently, the Ribeirão Preto EIP (Corrêa-Oliveira et al., 2022) has been implemented, featuring comparable interventions and methodology, including 237 first-episode patients between 2015 and 2018. While this program, like UNIFESP-EPM, is locally sustained, significant challenges regarding scale-up, access to care and integration in the mental health care system remain.

A randomized controlled trial has been conducted in Mexico comparing an early-psychosis-integrated program with standard care of pharmacotherapy alone. The program consisted of pharmacological and psychosocial treatment for patients, together with psychoeducation for relatives. Forty-four untreated FEP patients identified from the hospital of the National Institute of Psychiatry in Mexico City were recruited at baseline and followed for one year. Patients in the integrated program had shown improved outcomes regarding symptomatology, psychosocial functioning, lower rates of relapse and rehospitalization and high therapeutic adherence (Valencia et al., 2012). In Argentina, unlike the previously described programs in other countries that focus on providing support to the patients and their caregivers, an intervention was carried out targeting primary care health professionals (Padilla et al., 2015). The intervention aimed to reduce DUP in rural Argentina by providing primary care health workers with annual training that facilitates better screening and appropriate referrals (Padilla et al., 2015).

### Africa

Psychotic disorders account for a large proportion of years lived with disability within Africa (Whiteford et al., 2016). As in other parts of the world, problems of poverty, trauma and infectious diseases such as HIV and Malaria, all recognized risk factors for psychosis (Burns & Esterhuizen, 2008; Brown et al., 2020), are challenges in many regions; yet resources for providing evidence-based EIP care are almost universally inadequate and often inaccessible across the continent. This scarcity of provision of formal mental health services has led to high levels of unmet need. For example, the treatment gap for mental health care for people with psychosis has been shown in Ethiopia to be over 40%, while in

those receiving care, 72% were found not to have received minimally adequate care (Fekadu et al., 2019). Africa has the fewest mental health workers (0.9 per 100 000 population), mental health beds (2.5 per 100 000) and outpatient facilities (0.07 per 100 000) of any world region, while service users pay mostly or entirely out of pocket for treatment in 43% of African countries (World Health Organization, 2018). Most African countries have a dual system of care. Public healthcare is run by the government and largely free, and a private fee-for-service system is present.

Against this backdrop, to the best of our knowledge, there is not a single country on the African continent with a state or regional EIP in place (Hunt et al., 2022). Furthermore, evidence on interventions in FEP is scarce on the continent – there have been just a few trials completed or still in progress (Hunt et al., 2022). In addition, we are aware of a small number of individual localized initiatives providing specific FEP interventions within clinical programs; but none of these have published evaluations of these activities.

Given the lack of mental health services, a substantial part of the care burden falls on the shoulders of family members. A cross-sectional study from Tanzania is an example of research illustrating the issue of caregiver burden: they found that 63% of caregivers reported experiencing a high burden as a result of caring for a relative with schizophrenia (Clari et al., 2022).

It is widely recognized that a large proportion of individuals with early psychosis in African countries consult traditional and faith healers in their pathway to care (Burns & Tomita, 2015) and that such contact is associated with delays in accessing hospital treatment (i.e. long DUP; Burns & Tomita, 2015; Kaminga et al., 2020). This has motivated a number of studies exploring strategies to collaborate with traditional healers in detecting early psychosis in community settings (e.g. Morgan et al., 2015 and Gureje et al., 2020 in Nigeria and Ghana; and Veling et al., 2019 and Van der Zeijst et al., 2021 in South Africa) or to augment faith-healing facilities with psychopharmacological interventions (Ofori-Atta et al., 2017). The COSIMPO trial, a cluster-randomized trial of a manualized collaborative share care delivery intervention, delivered by trained traditional healers and primary health care providers, was conducted in Ibadan, Nigeria and Kumasi, Ghana (Gureje et al., 2020). At 6-month follow-up, a combination treatment of traditional healers and primary care workers was found to be more effective than enhanced care as usual at reducing psychotic symptoms and disability (Gureje et al., 2020). Individuals in both the intervention and control group experienced a significant decrease in harmful practices such as chaining (Gureje et al., 2020). Overall, collaboration between traditional healers and healthcare providers is likely beneficial for patients, despite perceived incompatibilities and mutual apprehensions regarding care strategies (Green & Colucci, 2020; van der Zeijst et al., 2023).

There are various initiatives that are not part of population-based or small-scale EIP programs but that can be considered preparatory activities for the development and implementation of such programs in the future. In Kampala, Uganda, for instance, a pilot randomized controlled trial of a psychoeducation intervention using trained village health team members aimed at improving treatment engagement and reducing symptoms in people with FEP, is currently in process (Akena et al., 2022). This stems from prior research by this group showing that the quality of individual and group-level interventions provided for people with FEP attending local services was poor (Mwesiga et al., 2021).

Other psychosocial initiatives not specific to FEP are nevertheless relevant to people with FEP and their families and caregivers. In Ethiopia, an RCT derived partly from the findings of PRIME (Hanlon et al., 2020), showed that "task-sharing" via training and supervision of local workers in the primary health care system was noninferior to specialized nurse care in a medical center (Hanlon et al., 2021). The patients in this RCT were drawn from a previous population-based study and had severe mental disorders including a large proportion with schizophrenia. In Malawi, a referral hotline and community mental health care team have been employed to increase awareness and referrals of individuals with psychosis overall, as well as to provide immediate treatment to individuals (Chilale et al., 2014; Kaminga et al., 2020). There have also been community-based psychosocial rehabilitation interventions (Brooke-Sumner et al., 2018; Asher et al., 2022) and a pilot RCT of a family intervention (Clari et al., 2022) for people with schizophrenia. There is also some preliminary work that indicated that multi-family psychoeducational groups might be acceptable to families of people with FEP in some urban settings (Asmal et al., 2014).

Clearly, there are significant research and clinical services gaps in relation to EIP within Africa. Three studies (INTREPID II in Nigeria, PSYMAP-ZN in South Africa and SCOPE in Ethiopia) are collecting evidence that will outline the epidemiology, risk factors and clinical presentation, course and outcome of FEP in African populations (Morgan et al., 2022; van der Zeijst et al., 2021; Asher et al., 2022). These studies will be a solid basis for clinicians and researchers across the continent to collaborate on redressing these EIP research and clinical services gaps. A key issue in developing EIP programs in Africa is the absolute necessity for such programs to be conceived within the local context. Thus, relevant issues such as how best to design effective task-sharing, how to collaborate with traditional healers, and how to ensure the culturally and socially acceptable participation of families and caregivers, must be considered from the outset. Of equal importance is the ethical principle of distributive justice – within a context where resources are limited, how do we go about providing EIP programs without drawing resources away from the few existing programs that do already exist for people with psychosis and other severe mental disorders?

## Discussion

In this narrative review, we provide an overview of population-based and small-scale, single-site programs for early psychosis in Africa, Asia and Latin America. There is a gradually growing number of single-site programs such as the SCARF program in Chennai, India, and UNIFESP-EPM in Sao Paulo, Brazil (Rangaswamy et al., 2012; Chaves, 2007), while population-based psychosis programs in these regions remain scarce. In addition to psychopharmacological care, most programs offer multicomponent, community-based mental health treatment tailored to the early phase of psychotic disorder including psychoeducation, employment and educational support, case management and family interventions.

### Shared challenges

An important challenge to appropriate treatment of mental health, particularly in LMICs, is stigma (Patel et al., 2018). Stigma and discrimination can be partly rooted in cultural factors, such as supernatural explanations for psychosis (Aliev et al., 2021; Makanjuola et al., 2016) and revolve around topics that are of cultural importance, like individuals appropriately filling social roles that

are viewed as normative by the local culture (e.g. roles of mother and wife for women; Angermeyer et al., 2016; Asher et al., 2018; Koschorke et al., 2014; Shrivastava et al., 2011). While anti-stigma interventions have shown promise (Mascayano et al., 2015; Maulik et al., 2016; Vaghee et al., 2015), stigma remains prevalent in virtually all areas, including family, community and mental health-care and can have impactful negative consequences such as reduced government spending on mental health and decreased willingness of individuals to seek professional help (Aliev et al., 2021; Angermeyer et al., 2016; Brenman et al., 2014; Gupta, Joshi, et al., 2021; Koschorke et al., 2014; Shrivastava et al., 2011). Even though stigma has been recognized as a barrier to care, understanding and reducing local manifestations of stigma and discrimination needs to remain a priority for LMICs to enable allocation of resources to provision of adequate care and establishment of EIP (Brenman et al., 2014; de Sousa et al., 2020; Gupta, Joshi, et al., 2021; Makhmud et al., 2022; Mascayano et al., 2015; Saraceno et al., 2007).

Another shared challenge across LMICs is the scarcity of epidemiological data regarding the incidence and prevalence of psychosis (Bastien et al., 2023). In the absence of high-quality data on the burden and distribution of psychosis, local governments will not be able to estimate the extent of the burden and need for services which is essential to begin planning EIP care. While there are various ongoing studies aimed at addressing this evidence gap, such as INTREPID (Morgan et al. 2022), a stark inequity in our knowledge of the epidemiology of psychosis in LMICs remains.

Additionally, in many LMICs, resources for mental health interventions remain very limited. It has been estimated that over 40 million people in LMICs need treatment for schizophrenia, with most countries having less than one psychiatrist available for a population of more than 100,000 people (Mari et al., 2009). While some progress has been made in recent years (World Health Organization, 2021), for example, Nepal increased available psychiatric beds by nearly sixfold (Rai et al., 2020), the lack of qualified psychiatrists (availability per 100,000 population: 0.1 in low-, 0.5 in lower-middle-, 2.1 in upper-middle-, 12.7 in HICs) and mental health workers in general (availability per 100,000 population: 1.6 in low-, 6.2 in lower-middle-, 20.6 in upper-middle- and 71.7 in HICs) in LMIC remains staggering (World Health Organization, 2018). Specifically, Africa remains most concerning in terms of mental health resources at 0.9 mental health workers and 2.5 mental health beds per 100,000 people (World Health Organization, 2018). This lack of personnel is likely responsible for concerning findings such as the 89% and 69% treatment gap for psychotic disorders in low-income and low-middle-income countries respectively (Lora et al., 2011), which has been corroborated in recent years (e.g. 75.5% treatment gap for psychotic disorders in India; Gautham et al., 2020).

Although Global Mental Health is a field that often faces challenges with regard to access to resources and funding, it is important to acknowledge a more optimistic perspective that exists within the field. This perspective recognizes the potential of community resources and local experiences in adapting and implementing evidence-based practices. Despite the lack of resources and financial gaps that may exist, many communities have valuable assets that can be leveraged to support mental health interventions. These assets may include traditional healing practices, social support networks, including an active role of families in care provision and community leaders who can act as advocates for mental health. By recognizing and building upon these assets, mental health practitioners can work collaboratively with communities to develop and implement culturally relevant and effective interventions. This

approach can also help to overcome some of the barriers that exist in accessing traditional mental health services, such as stigma and lack of trust toward and among people with lived experience.

### Distribution of resources

In contrast with the lack of dedicated EIP programs in many regions, some East Asian HICs, including Singapore and Hong Kong, have publicly funded mental health systems. Their population-based EIP programs are well-established and underpinned by data acquired from participants demonstrating the beneficial effects of EIP programs on short- and longer-term functional and symptomatic outcomes (e.g., Hui et al., 2018; Chan et al., 2018, 2020). In other countries with a fee-for-service or private mental health services such as in large parts of Japan, EIP programs may not be accessible to disadvantaged communities. This selective overview demonstrates how HICs and LMICs in similar regions vary in their availability of EIP programs. (O'Connell et al., 2021) This indicates that equitable delivery of EIP programs is not only dependent on available resources but also on funding priorities set by lawmakers.

The lack of resources in many regions combined with supportive evidence of early identification of psychosis, calls for locally-led, culturally adapted, collaborative community-based interventions for people with early psychosis (Naeem et al., 2015). Meta-analytic evidence on community-based psychological interventions for schizophrenia in resource-strapped settings has indicated beneficial effects on symptom severity and hospital readmissions (Asher et al., 2018). Given research suggesting alarmingly high mortality among people with psychotic disorder in LMICs (Cohen, 2023), for a large part driven by poor physical health, this should be integrated as a key target in the implementation of EIP programs. Overall, more high-quality research is needed to demonstrate the effectiveness and cost-effectiveness of multicomponent EIP programs in LMICs.

Importantly, there is also criticism of the early intervention model, especially the concentration of resources in the early stages of psychosis (Aceituno et al., 2020; Keshavan et al., 2010). This may lead to a shortage of resources at later stages of psychotic illness. In the context of LMICs which, in most cases, already have to deal with a grossly under-resourced mental health system, this may increase disparities in access to and the delivery of mental health care. It is therefore important that investments are made in capacity building to enable effective implementation of EIP which can be scaled up to the extent allowed by the resources available in the local mental health system.

### Task-sharing and integrated primary healthcare

One strategy to address the treatment gap of psychotic disorder in LMICs may involve task-sharing. Task-sharing is the process in which psychological interventions are carried out by less specialized staff or lay health workers to increase the capacity and coverage of mental health services in resource-strapped settings. There have also been several initiatives aimed at training primary healthcare workers in providing mental health interventions to people with psychotic disorders, including schizophrenia. Some initial findings from the Community Care for People with Schizophrenia (COPSI) study, carried out across three sites in India, indicate that multicomponent community-based care delivered by trained lay health workers including psychoeducation, rehabilitation and health promotion was acceptable and feasible for people with schizophrenia (Balaji et al., 2012). In Ethiopia, task-sharing has been shown to be

noninferior in integrated mental health care for individuals with severe mental disorders, including psychosis (Hanlon et al., 2021). Similarly, the Rehabilitation Intervention for people with schizophrenia in Ethiopia (RISE; Asher et al., 2022) study also provided evidence that community-based rehabilitation services delivered through a stepped care model including task-sharing could be effective in reducing caregiver-rated disability of people with schizophrenia. Of note, however, these interventions have been designed to treat schizophrenia, while early psychosis programs include interventions specifically designed to target people in the early stage of the illness. There is, however, promising evidence in this regard, for example, lay health workers were trained and provided mental health services to young (aged 14–30 years) people with major mental disorders (including psychosis) in the conflict-ridden region of Kashmir in India with no formal services (Malla et al., 2019). A significantly high number of patients were identified and treated during the study; substantial clinical, functional and quality-of-life improvements were noted, with high levels of treatment engagement (Malla et al., 2019). Further research should test this type of intervention for addressing the needs of persons with early psychosis in low-resource settings.

In addition, the involvement of people with lived experience is gaining prominence in EIP treatment programs. Over the past decades, people with lived experience have successfully organized and advocated for improved mental health services; peer-operated alternatives; and much greater inclusion in national and local mental health policy and planning initiatives, governance and administration. Their involvement in task-sharing could be an opportunity to invest in and empower people with lived experience so that they can carry out some of the mental health support services that are so direly needed in under-resourced settings.

In the discussion of our findings, several limitations should be addressed. First, we could only provide a snapshot of the current situation regarding EIP programs in the Global South. New programs appear, existing programs evolve regarding the services they provide or the population they serve, while other programs cease to exist, for example, because of a lack of financial investment. Second, there is a large amount of heterogeneity across programs in the services they provide which compromises their comparability. Moreover, information on some programs was limited which made it difficult to judge whether programs could be considered EIP programs, or for instance, programs only aimed at improving the detection of FEP.

In conclusion, the uptake of EIP programs in countries with developing economies and LMICs is extremely slow. Most existing programs are small-scale, single-site programs, while in Chile and China, efforts are made to implement population-based programs for the detection and treatment of people with psychosis. Quantitative and qualitative data are needed to learn more about the needs of people with FEP and their families, as well as contextual factors that predict successful implementation and cultural buy-in.

**Open peer review.** To view the open peer review materials for this article, please visit http://doi.org/10.1017/gmh.2024.78.

**Supplementary material.** The supplementary material for this article can be found at http://doi.org/10.1017/gmh.2024.78.

**Data availability statement.** No original data has been collected for this manuscript.

**Acknowledgments.** We are thankful to Sophie Blackmore, BSc, and Aarati Taksal, PhD, for their help with the references of this manuscript.

**Author contribution statement.** Conception and design (EvdV, ES); drafting (EvdV, XY, FM, KJW, EC, CYZT, SWK, JB, BC, GM, SI, TR; revising the manuscript critically for important intellectual content (all)

**Financial support.** This research received no specific grant from any funding agency, commercial or not-for-profit sectors.

**Competing interest.** None.

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
