## [Reviewer Report]

Review

Title: Early Intervention in Psychosis Programs in Africa, Asia and South America; Challenges and recommendations

This narrative review focus on the Africa, Asia and South American regions. It covers quite comprehensively the LMIC of the Southern part of the globe. Author have described results mainly in the context of these areas. To improve further in terms of fitting the results with global learning, I wonder if authors could mention a little bit on some of the shared challenges of the regions and other parts of the world such as stigma and lack of resources etc.

Though this is a narrative review, methods of article search should be reported in more detailed. These would include which electronic search engine was selected etc.

This review has already covered quite substantially the EIP programs of Africa, Asia and South America with relevant and interesting conclusions. In page 12, line 25-26, should add reference Chan S.K.W., Chau E.H.S., Hui C.L.M., Chang W.C., Lee E.H.M. and Chen E.Y.H. (2016). Long term effect of early intervention service on duration of untreated psychosis in youth and adult population in Hong Kong. Early Intervention in Psychiatry, 12(3): 331-338. doi: 10.1111/eip.12313. This article outlined the detailed community awareness program and its impact on DUP reduction. On the same page, in line 48-49, should add reference Lau K.W., Chan S.K.W., Hui C.L.M., Lee E.H.M., Chang W.C., Chong C.S.Y., Lo W.T.L. and Chen E.Y.H. (2017). Rates and predictors of disengagement of patients with first-episode psychosis from the Early Intervention Service for Psychosis Service (EASY) covering 15 to 64 years of age in Hong Kong. Early Intervention in Psychiatry, 13(3): 398-404. doi: 10.1111/eip.12491. This article describes the extended EASY service.

---

## [Reviewer Report]

Early intervention in psychosis programs in Africa, Asia and South America; challenges and recommendations

Overall, major points:

1. The authors take a useful approach of considering world regions in the Global South (Africa, Asia, Latin America), with comparison between countries of differing economic status within those regions. It is relevant to focus on the Global South rather than a fully global review (including the Global North) because of the interest in understanding how established approaches to Early Intervention for Psychosis programmes in the Global North may need to be rethought across diverse Global South settings.

2. Although an illuminating paper with potential to be helpful in developing this neglected area of global mental health, the description of this paper as a ‘narrative review’ seems incorrect. It seems to be rather an expert group that is drawing on their own networks and links to describe EIP programmes and lessons learned, rather than a systematic attempt to map out programmes and associated evidence. The paper would be better (more accurately) framed as a commentary.

3. There is something of a conceptual muddle between EIP programmes and population-based programmes seeking to expand access to mental health care in general (not just ‘early’ or even ‘first contact’). As the conceptual approach has not been applied systematically, important examples of population-based efforts to expand access to mental health care for people with psychosis have been missed – China 686 programme is in, but (for example) the PRIME models of care in Nepal and Ethiopia are not included. This would also help the paper by bringing in examples of ‘task-shared’ care (better referred to as integrated primary mental health care) that can be used to achieve earlier care tailored to needs (this gets picked up in the discussion but doesn’t actually refer to existing examples).

4. There is also some muddle about, for example, the relevance of traditional and religious healing, stigma and differing roles of the family as part of an EIP response. This is not region-specific and the discussion of these important issues rather need to be cross-cutting.

5. A key concern about transporting EIP models from the Global North is in terms of the specialised human resources upon which they depend. At the moment this commentary neglects to provide detail of the personnel delivering EIP in diverse Global South settings. Adding this to the table or the more in-depth descriptions of the EIP models that the authors know well would be beneficial.

6. Although the authors mention the high mortality in the SCARF-McGill collaborative study, echoing findings from other studies in Africa, there is nothing about how this might need to be a key focus of EIP services. In the Ethiopia studies, mortality was driven by poor physical health/healthcare, with evidence that provision of mental health care alone did not ameliorate the excess risk.

Specific points:

Abstract

7. Please add details of the review methodology. It seems as if it was purposive, informed by experts?

8. May be relevant to note that ‘first contact’ rather than ‘early intervention’ could be more relevant in settings with low treatment coverage.

9. Could also be worth noting in the abstract whether there has been any attempt to integrate early intervention into primary care settings and reliant on non-psychiatrist mental health professionals?

Methods

10. Please note which databases were searched.

Results

11. What research evidence or grey literature evidence was available?

12. It would be helpful to introduce the results by stating the number of EIP programmes identified and included.

13. Table 1 – is it possible to add the human resources involved in delivering EIP services?

14. The 686 is not an early intervention programme, as recognised by the authors. Rather it is an attempt to increase population coverage of basic mental health care, especially for people with psychosis. This distinction is recognised by the authors, but if you include 686 then you also need to include other population-based efforts to expand access to mental health care e.g. the Programme for Improving Mental health carE for people with psychosis in Nepal and Ethiopia.

15. Page 15 (South Asia). The following is a rather general statement which seems unhelpful given the diversity of cultures within these settings (“Another common thread binding the South Asian countries are some of their similar values and norms followed. While the concept of culture is multilayered, influenced by factors like language and nationality amongst others, mental health is deeply rooted in all holistic aspects of social constructs like shared history, religion, and family values”). Any such assertion would need to be better substantiated.

16. On page 15/16 the comments about the irrelevance of HIC models of EIP seems relevant to all low-income country settings, not just those in South Asia. The meaning of the reference to colonial history and oppression is not clear. Which countries are being referred to and what are the specific legacies that are problematic for EIP development? It is a strength to make reference to such histories, but the argument would be stronger if the points are more specific and evidenced.

17. SCARF is a WHO collaborating centre rather than ‘research centre’.

18. There is a strong focus on the EIP service in Chennai. Given the depth of data about that service, it would be helpful to know more about who delivers the programme (same mix of specialists as in HICs or any degree of task-shared care?) and how this programme fits with other mental health care – is it a first contact or early intervention programme? How do people ‘graduate’ to other mental health care? How is the mean DUP so short? (are there community-based activities or does this reflect that SCARF is a special case because of their exemplary work over many years in these communities?). How scalable is this model?

19. In the Pakistan/Nepal section, reference is made to collaborative models with traditional and religious healers, but this point is more widely applicable to other geographies. Furthermore, the citation is for the COSIMPO trial which took place in Nigeria and Ghana. It would be highly relevant to mention the Nepal PRIME study and the efforts of that team to increase early detection of psychosis in the community (the ‘CIDD’ – Mark Jordans, Nagendra Luitel et al.).

20. In the section of Chile, it is relevant to emphasise that it is a high-income country, albeit an ‘emerging economy’. The authors have some reflections on the cultural fit of the US model that is being trialled. Could they elaborate further? How is the national programme seeking to reach under-served populations? How well does Western concepts of ‘recovery’ fit?

21. The Argentina example is the first to mention primary care but seems to be about primary care as a point of referral but not of treatment.

22. The first few sentences of the paragraph on Africa are general and could equally apply to other world regions. For example, is substance use higher in Africa than the US? (I doubt). Trauma is an issue in some African countries, but not in many others, and is also prominent as an issue in other continents. Are years lived with disability higher in Africa? The wording somewhat reinforces deficit-models of ‘Africa’ and risks ignoring diversity and cultural richness. I suggest re-writing and emphasising the lack of provision of formal mental health services leading to high levels of unmet need.

23. It is not clear why a single small study on caregiver burden from Tanzania is singled out for citation when there have been numerous studies on the continent.

24. It is relevant to mention the other RCT of a collaborative model of care with prayer camps in Ghana (Ofori-Atta et al).

25. It is also relevant to mention PRIME Ethiopia, that achieved high levels of mental health service (in primary care) coverage for people with psychosis in a defined geographical area. This model had a community component to proactively identify people with psychosis in the community.

26. In what specific ways can ‘unity be strength’? Sounds like a nice idea, but what does this need to look like and why is this just relevant for Africa?

Discussion

27. When mentioning COPSI, it is also relevant to mention the RISE trial (Asher et al) in Ethiopia which had a similar community-based rehabilitation model for people with psychosis but linked to mental health care integrated in primary care (delivered by general health workers).